# Unlocking Tokens as Data Points for Generalization Bounds on Larger Language Models

**Sanae Lotfi**[1,*]     **Yilun Kuang**[1,*]     **Brandon Amos**[2,†]

**Micah Goldblum**[3]     **Marc Finzi**[4]     **Andrew Gordon Wilson**[1]

[1]New York University   [2]Meta AI   [3]Columbia University   [4]Carnegie Mellon University

## Abstract

Large language models (LLMs) with billions of parameters excel at predicting the next token in a sequence. Recent work computes non-vacuous compression-based generalization bounds for LLMs, but these bounds are vacuous for large models at the billion-parameter scale. Moreover, these bounds are obtained through restrictive compression techniques, bounding compressed models that generate low-quality text. Additionally, the tightness of these existing bounds depends on the number of IID documents in a training set rather than the much larger number of non-IID constituent tokens, leaving untapped potential for tighter bounds. In this work, we instead use properties of martingales to derive generalization bounds that benefit from the vast number of tokens in LLM training sets. Since a dataset contains far more tokens than documents, our generalization bounds not only tolerate but actually benefit from far less restrictive compression schemes. With Monarch matrices, Kronecker factorizations, and post-training quantization, we achieve non-vacuous generalization bounds for LLMs as large as LLaMA2-70B. Unlike previous approaches, our work achieves the first non-vacuous bounds for models that are deployed in practice and generate high-quality text.

## 1   Introduction

Despite the impressive empirical performance of large language models (LLMs), our theoretical understanding of their performance is lacking. PAC-Bayes and the related finite hypothesis generalization bounds [5, 13, 17] offer a compelling framework for understanding this good performance through the lens of compression. These bounds tell us that a model will provide good generalization if it is capable of fitting its training data while simultaneously being compressible relative to the size of its training set. The generalization bounds literature includes many techniques for achieving tighter bounds on image classification problems, ranging from improved bounds themselves to new compression methods [53, 14, 18, 38, 31].

Recent work presented the first non-vacuous generalization bounds for large language models, considering training points to be independent and identically distributed (IID) documents [32]. The authors compute generalization bounds for the expected bits-per-dimension (BPD) loss, defined for a document $X$ composed of $k$ tokens and a language model $h$ as the average negative log probability $\mathrm{BPD}(h, X) = -\frac{1}{k} \sum_i^k \log_2 p_h(x_i | x_{<i})$. These bounds are only non-vacuous for compressed

---

*Equal contribution, order decided by coin flip. Correspondence to: Sanae Lotfi <sl8160@nyu.edu>, Yilun Kuang <yilun.kuang@nyu.edu>, Andrew Gordon Wilson <andrewgw@cims.nyu.edu>.

†Meta-affiliated author was involved only in an advisory role. All experimentation and data processing were conducted at NYU.

38th Conference on Neural Information Processing Systems (NeurIPS 2024).

GPT2 variants [39] that output un-grammatical text. The term *vacuous* refers to the random guess performance on next token prediction, which is $\log_2 V$ for BPD where $V$ is the vocabulary size.

Compression-based generalization bounds at the document level suffer from three primary limitations: (1) the number of documents in a training set is limited, and this small sample size leads to loose bounds; (2) due to the small sample size, non-vacuous generalization bounds can only be achieved using compression techniques which significantly modify the LLM pretraining routine. This limitation also applies to state-of-the-art generalization bounds for image classification, which heavily alter the training procedure to optimize the bounds [53, 38, 31]; (3) as a result, the models which produce non-vacuous bounds generate low-quality text, so it is unclear what these bounds can tell us about more performant language models.

In this work, we address the above limitations and use our bounds to derive insights about the generalization properties and limitations of LLMs. Namely, we make the following contributions:

- In Section 4, we derive a new generalization bound that considers each sample to be an individual token. Even though tokens within a document are not independent, we use properties of martingales to obtain a valid bound that benefits from the number of tokens in a language model's pretraining dataset.

- In Sections 5 and 6, we explore several expressive model compression techniques such as Monarch matrices, Kronecker factorizations, and post-training quantization and show that bounding the performance at the token-level favors less restrictive compression strategies.

- Our work is the first to compute non-vacuous generalization bounds for models compressed only through post-training quantization and without altering the pretraining procedure at all. Consequently, we obtain generalization bounds for massive pretrained LLMs like LLaMA2-70B, as shown in Figure 1(Left) and Section 6, which generate high-quality text.

- Our experiments in Section 6 indicate that the chat versions of LLaMA have looser generalization guarantees, demonstrating that fine-tuning these models for dialogue negatively affects their performance on the next token prediction task.

- In Section 6.4, we demonstrate that GPT2 models that are restricted to only seeing $k$ tokens in their context for training and evaluation obtain significantly better bounds than $k$-th order Markov chains for high values of $k$, reflecting the remarkable ability of transformer-based models in capturing longer range correlations.

- We show in Section 6.5 that a model's ability to recall memorized facts from its pretraining data deteriorates faster than its ability to recognize structured patterns as we decrease the size of the model through compression, distinguishing between compressible tasks where generalization is possible and incompressible tasks that correspond to sheer memorization.

We make our code underline{available here}.

## 2   Background

In this section, we review the different components of compression-based generalization bounds, which we build upon with our method in Sections 4 and 5.

**Finite hypothesis compression bounds.** Let $R(h, x) \in [a, a + \Delta]$ be a bounded risk and $h \in \mathcal{H}$ be a hypothesis drawn from a finite hypothesis space with prior $P(h)$. A classic finite hypothesis generalization bound [43] states that for any $\delta > 0$ with probability $1 - \delta$,

$$R(h) \leq \hat{R}(h) + \Delta \sqrt{\frac{\log 1/P(h) + \log 1/\delta}{2m}} \tag{1}$$

where the empirical risk is defined as $\hat{R}(h) := \frac{1}{m} \sum_{i=1}^{m} R(h, x_i)$ with $\{x_i\}_{i=1}^{m}$ being IID and $R(h) = \mathbb{E}[\hat{R}(h)]$. The complexity term depends on the prior log probability $\log 1/P(h)$. We use the Solomonoff prior $P(h) \leq 2^{-K(h)}$ [45], where $K(h)$ is the prefix Kolmogorov complexity of $h$ defined as the length of the shortest program that produces $h$ for a fixed programming language [23]. Consequently, our prior favors models $h$ that have a small minimum compressed length. While the Kolmogorov complexity is incomputable, it can be bounded as $\log 1/P(h) \leq K(h) \log 2 \leq C(h) \log 2 + 2 \log C(h)$, where $C(h)$ is the compressed size of the model according to a pre-specified

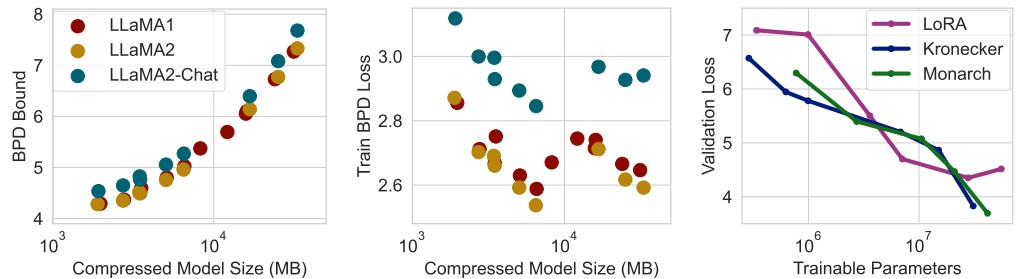

Figure 1: **Non-vacuous bounds for LLMs that scale up to 70B parameters. Left:** Bits per dimension (BPD) bounds on the Amber dataset [29] which contains 1.2 trillion tokens for different LLMs from the LLaMA family ranging in scale from 7 billion to 70 billion parameters [47]. All of these models are quantized to 2-bits, 3-bits and 4-bits per-weight using QuIP# and are publicly available [48]. The different quantization precisions are accounted for in the compressed model size. The trade-off between the empirical performance and the model complexity in our bounds favors models with a smaller compressed size in general, though we observe that across different architectures we can find larger models yielding better bounds. **Middle:** The BPD training loss for different models from the LLaMA family—the legend is shared with the figure on the left. Overall, we observe that larger models yield a lower BPD while having a higher compressed size. **Right:** Validation negative log-likelihood loss as a function of the total number of trainable parameters for different nonlinear parametrization; namely low rank adaptation (LoRA), the Kronecker decomposition of dense matrices and Monarch matrices. The x-axis is in the log scale. As we vary the number of trainable parameters, there are different optimal compression techniques.

compressor. Therefore, we can find the right trade-off between the empirical risk and the compressed size of the model by tuning the extent of compression, hence the different compression techniques we explore in this work.

**Compression bounds for LLMs.** When constructing document-level bounds for language, the empirical risk is defined over an entire document $X$ as $R(h, X) = -\log_2 p_h(X)/L$, where $p_h(X)$ is defined auto-regressively on the sequence of tokens $X = [x_1, x_2, \ldots x_L]$ as $p_\theta(X) = \prod_{i=1}^{L} p_h(x_i|x_{<i})$, where $x_{<i}$ denotes $x_1, x_2, \ldots, x_{i-1}$.

**Prediction smoothing.** Since the bound in Equation (1) only applies to a bounded risk, it is not valid for the bits-per-dimension loss that is unbounded. In this case, one can introduce a prediction smoothing probability $\alpha$ to the predictive model such that the generative probability distribution becomes a mixture between the next token probability according to the auto-regressive model $f(\theta)$ with parameters $\theta$ and a uniform distribution over the vocabulary of size $V$ as follows: $p_h(x_i|x_{<i}) = (1 - \alpha)p_\theta(x_i|x_{<i}) + \alpha/V$. With this construction, $R(h, X)$ can be bounded in an interval of size $\Delta = \log_2(1 + (1 - \alpha)V/\alpha)$. The optimal hyperparameter $\alpha$ is determined via a grid search in Lotfi et al. [32].

**Compressing LLMs with SubLoRA.** To achieve the extreme compression level necessary to obtain non-vacuous document-level bounds, Lotfi et al. [32] propose SubLoRA, a non-linear subspace parametrization of an LLM's weights $\theta$. Using SubLoRA, these weights can be written as $\theta = \theta_0 + \text{LoRA}(Pw)$. Here $\theta_0 \in \mathbb{R}^D$ are the model weights at random initialization and $\text{LoRA}(Pw)$ combines low-rank adaptation (LoRA) [19] with subspace training [31] via the projector $P \in \mathbb{R}^{D \times d}$. The LoRA decomposition parameterizes a dense matrix $W \in \mathbb{R}^{a \times b}$ as the product of two low-rank matrices $A \in \mathbb{R}^{a \times r}, B \in \mathbb{R}^{r \times b}$ with a small rank $r$. As for the linear subspace parametrization $Pw$, the projection matrix $P$ is defined as a Kronecker product $P = Q_1 \otimes Q_2$ produced by orthogonalizing $Q_1, Q_2 \sim \mathcal{N}(0, 1/\sqrt{D})^{\sqrt{D} \times \sqrt{d}}$ via a QR decomposition.

In practice, a selected subset of the dense matrices in an LLM are parameterized using LoRA's low rank matrices, then the concatenation of LoRA's matrices is projected into the subspace parameters $w$ using $P$. The model is therefore effectively trained via the weights $w \in \mathbb{R}^d$. As a result, the model can be coded via a random seed that reproduces the pre-fixed initialization $\theta_0$ and projection matrix $P$, and a coding of $w$ which is performed using arithmetic coding [25]. The dimension $d$ of $w$ can

be varied to achieve the best trade-off between empirical risk and complexity, and these degrees of freedom are accounted for in the coding of the hypothesis $h$.

## 3 Related Work

**Generalization bounds for neural networks.** Deep neural networks are challenging to understand using generalization theory due to their many parameters [51]. However, over the past years, there has been success in constructing meaningful bounds covering for image classification models [13], vision-language models [1], and tabular data [17], often through the methodology of compression [53, 31]. Lotfi et al. [32] extend compression-based generalization bounds to the LLM setting, and obtain non-vacuous bounds at the document level. Li et al. [27] explore generalization in few-shot learning, establishing bounds based on in-context examples while maintaining a fixed pretrained model. In contrast, we investigate pretraining generalization bounds to understand why models do not overfit at training time, despite the increased dataset complexity.

**Non-IID Generalization bounds.** Ralaivola et al. [41] analyze the dependence graph of the random variables, deriving a bound based on the graph coloring number, fitting into a broader line of work making use of properties of the dependence graph [52]. Unfortunately for text data, the dependencies are unknown or assumed to follow the triangular autoregressive dependency structure for all pairs in the sequence. A related line of work has been to explicitly estimate coefficients which quantify the extent that random variables relate to each other, [e.g., 33, 24]. However, it is unclear how best to apply these methods to neural networks. Martingale tail bounds are sometimes used in online learning and reinforcement learning, e.g., for establishing regret bounds [40]. Chugg et al. [7] present a large collection of generalization bounds both in the IID and martingale settings, including generalization bounds which could be used at the token level such as the one we derive. Their results extend and generalize many existing bounds. We view our contribution as orthogonal to these efforts since we focus on constructing the components necessary to generate practical bounds for LLMs, rather than abstractly innovating on concentration inequalities.

**Large language models and compression.** Parameter-efficient finetuning methods, such as LoRA [19], parametrize weight matrices as products of two trainable low-rank matrices on top of frozen pretrained weights. QLoRA uses 4-bit NormalFloat (NF4) and double quantization, enabling single-GPU finetuning for a 65B parameter LLM without performance degradation [10, 11]. Post-training quantization approaches, such as GPTQ [16], rely on second-order information and quantize each row of weight matrices independently. QuIP uses adaptive rounding and incoherence processing of second-order Hessian matrices, enabling 2-bit quantization of LLMs [6]. Other compression techniques for LLMs include replacing most of the 16-bit operations with 8-bit matrix multiply [10], using data-free distillations [28], designing custom kernels and sub-4-bit integer quantization [22, 36], and compressing embeddings as low-rank matrix-product state [50].

## 4 Token-Level Generalization Bounds

In order to unlock a deeper understanding of LLM generalization, it is not sufficient to consider the training data at the level of entire documents. In fact, token-level performance is arguably what we care about most when evaluating a model's generalization on its next token prediction pretraining task. Moreover, simplifying the bounds to meet the IID assumption over sampled documents restricts our ability to capture the dependencies between individual tokens. In this section, we derive novel bounds at the token level through a simple yet powerful application of Azuma's inequality that allows us to use the properties of martingales to go beyond the IID setting. Then, we discuss the interpretation of our bounds and demonstrate their ability to predict downstream generalization. Finally, we introduce a new optimization strategy for tuning the prediction smoothing hyperparameter.

### 4.1 A Novel Non-IID Token-Level Generalization Bound

In deriving token-level bounds, one might consider applying Equation (1) to the finite dataset $\mathcal{D} = \{(x_{<i}, x_i)\}_{i=1}^{M}$ composed of input and output pairs. In this scenario, model training can be performed on a random subset $S \subset \mathcal{D}$ of $m$ pairs, which differs from how training is usually performed via contiguous sequences. Then, we could use the performance on $S$ to bound the average performance on $\mathcal{D}$ since $S$ is constructed as an IID sample from $\mathcal{D}$. While these bounds are valid,

they require fundamentally altering the training procedure, and they only pertain to the held out pairs which must be collected in advance and separated from their naturally occurring context.

To avoid these limitations, we construct a novel bound that naturally accommodates the non-IID structure of the tokens as they occur in documents as follows:

**Theorem 4.1.** *With probability at least $1 - \delta$ over the randomness in a sampled sequence $\{x_1, x_2, \ldots, x_m\}$, if the negative log likelihood of a model $h \in \mathcal{H}$ can be bounded $-\log_2 p_h(\cdot|x_{<i}) \in [a, a + \Delta_i]$, then the negative log likelihood of the data for model $h$ satisfies*

$$\frac{1}{m} \sum_{i=1}^{m} \mathbb{E}[-\log_2 p_h(X_i|x_{<i})|x_{<i}] \leq -\frac{1}{m} \log_2 p_h(x_{\leq m}) + \hat{\Delta}\sqrt{\frac{\log 1/P(h) + \log 1/\delta}{2m}}, \quad (2)$$

*where $\hat{\Delta} = \sqrt{\frac{1}{m}\sum_{i=1}^{m} \Delta_i^2}$, the expectation is taken over $X_i \sim p(X_i|x_{<i})$ from the data generating process, and $P(h)$ is any normalized prior over a discrete hypothesis space $\mathcal{H}$ that does not depend on $\{x_i\}_{i=1}^m$.*

We provide a proof sketch as well as the full proof in Appendix A.1.

On the right-hand side of the bound is the conventional empirical risk: $-\frac{1}{m} \log_2 p_h(x_{\leq m}) = -\frac{1}{m} \sum_i \log_2 p_h(x_i|x_{<i})$ on the measured sequence and a complexity term $\log 1/P(h)$. We describe in detail how we sample sequence $x_{\leq m}$ and compute the empirical risk in Section 4.2. The quantity which we are bounding on the left-hand side is the expected next token negative log-likelihood under resampling from the data generating process, averaged over the different contexts that have been encountered in the training set. The bound ensures generalization on contexts seen at training when the next tokens are resampled, but not on data with contexts that are different. However, given how diffuse the distribution over next tokens is, e.g., at the beginning of a new sentence, our bounds remain predictive of generalization and achieving a non-vacuous bound requires generalization. We provide further interpretation of the bounds, including a protein application, in Section 6.

## 4.2 Sampling and Empirical Risk Evaluation

In this section, we more precisely define the sequence $x_{\leq m}$ for which we compute the empirical risk in Equation (2). We construct a sample $x_{\leq m}$ from the stochastic process $p_{\text{data}}$ by first sampling independent and identically distributed documents, e.g., the documents that form the OpenWebText dataset. Then, we concatenate these documents deterministically using end of text (EOT) tokens. Consequently, the ground truth stochastic process has the following property:

$$p_{\text{data}}(x_i|x_{<i}) = p_{\text{data}}(x_i|x_k, \ldots, x_{i-1}), \quad (3)$$

where $x_k$ is the previous EOT token. This equality holds exactly due to how the stochastic process is implemented.

On the other hand, it would not be guaranteed that a generative model $p_h(x)$ satisfies the property in Equation (3) apriori if the model were allowed to attend to tokens $x_{<k}$, even when the data generating process has this property. However, we explicitly prohibit our generative model $h$ from attending to tokens $x_{<k}$ through the attention mask, as we have the flexibility to do so in defining our hypothesis class and model family. Therefore, our model $p_h$ that we bound also satisfies this property $p_h(x_i|x_{<i}) = p_h(x_i|x_k, \ldots, x_{i-1})$ exactly, and not approximately.

In conclusion, the empirical risk for our generative model $h$ and a sequence $x_{\leq m}$ sampled from the stochastic process defined above can be written as follows:

$$-\frac{1}{m} \log_2 p_h(x_{\leq m}) = -\frac{1}{m} \sum_i \log_2 p_h(x_i|x_{<i}) = -\frac{1}{m} \sum_i \log_2 p_h(x_i|x_k, \ldots x_{i-1}),$$

where $x_k$ is the nearest EOT token occurring before $x_i$. Given the large size of the OpenWebText and Amber datasets, containing 9 billions and 1.2 trillion tokens respectively, we use subsampling for the evaluation of the empirical risk. More details can be found in Appendix A.2.

## 4.3 Token-level Bounds Are Predictive of Generalization

**Token-level vs. document-level bounds.** In contrast to document-level bounds, our token-level bounds increase the number of samples, driving down the size of the complexity term, and do

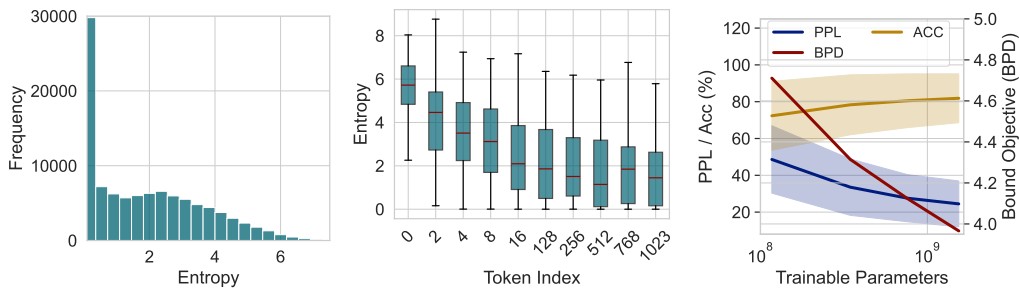

Figure 2: **Our bounds analyze a quantity that is meaningful and predictive of generalization.**
**Left:** Using LLaMA2-7B, we compute the entropy of $p(x_i|x_{<i})$, where the context $x_{<i}$ is fixed and sampled from the Amber training dataset. The distribution over next tokens given a fixed context from the training data is indeed diffuse and characterized by high entropy values. **Middle:** Entropy of $p(x_i|x_{<i})$ as a function of the token index $i$ shown on the x-axis for a context length $L = 1024$. The average entropy has a decreasing trend but remains high overall; note that the average entropy for $i = 768$ is as high as the average entropy for $i = 128$. **Right:** On the left $y$-axis, we plot the average zero-shot accuracy (ACC) and perplexity (PPL) achieved by GPT2 models ranging in scale from 117M to 1.5B averaged over downstream datasets, as reported in Radford et al. [39]. On the right $y$-axis, we plot an approximation of the conditional BPD expectation that we bound in Equation (2) where we resample $x_i$ from a LLaMA2-7B given fixed training contexts $x_{<i}$ from the Amber dataset. The approximation of the BPD objective that we bound achieves 97.9% and 99.1% correlation with the accuracy and perplexity, respectively.

not require the IID assumption. Whereas the number of samples previously would be the number of documents, it is now simply the number of tokens in the dataset, a far higher number. As a consequence of decreasing the complexity term, the empirical risk will be a more significant contributor to our bounds compared to document-level bounds. Therefore, we achieve non-vacuous bounds for much larger and more performant models that generate high-quality text. This development brings our theoretical bounds much closer to aligning with empirical generalization.

**Interpretation of token-level bounds.** It is important to note the difference between the quantity that we bound $\frac{1}{m}\sum_{i=1}^{m}\mathbb{E}[-\log_2 p_h(X_i|x_{<i})|x_{<i}]$, which is conditioned on contexts seen at training, and the expected risk $\mathbb{E}[-\log_2 p_h(X_i|x_{<i})]$ under resampling from the data generating process where new contexts can be sampled from this process. However, the resampled next tokens $x_i|x_{<i}$ are not necessarily from the training set, and to the extent that the distribution over next tokens is entropic, we are measuring a different quantity than the empirical training performance of the hypothesis $h$. Moreover, we know that the distribution over next tokens is often indeed diffuse; for instance, many words have common synonyms. The distribution over next tokens is especially diffuse when we start a new sentence, for example. We demonstrate how diffuse the distribution $p(x_i|x_{<i})$ is for fixed contexts $x_{<i}$ from the publicly available Amber training dataset [29] (see Appendix B.7) by sampling $x_i|x_{<i}$ using LLaMA2-7B to approximate the generative process. Figure 2(Left) shows that, indeed, the distribution $p(x_i|x_{<i})$ is characterized by a high entropy for a large number of tokens. In Figure 2(Middle), we plot the entropy of $p(x_i|x_{<i})$ for each index $i$ in a context of length 1024. This figure confirms our intuition that the next token distribution is particularly diffuse at the beginning of a sentence, while it decreases for later tokens but remains relatively high. Given how diffuse the distribution is and the large number of possible sentences, it is broadly infeasible to make predictions on new resampled tokens from the empirical distribution alone.

**Our bounds are predictive of downstream performance.** We compute an approximation of the quantity that we bound in Equation (2) by sampling next tokens $x_i$ using LLaMA2-7B given fixed contexts $x_{<i}$ from the Amber dataset. We plot this quantity on the right $y$-axis of Figure 2(Right), and show on the left $y$-axis the performance of GPT2 models of varying sizes on downstream datasets as reported in Radford et al. [39]; see Appendix B.4 for more details. Not only does the approximation of the BPD objective show the same trend as the downstream performance for different GPT2 variants, but it also achieves 97.9% and 99.1% correlation [4] with downstream task accuracy and perplexity metrics, respectively. Moreover, we show in Appendix C.3 that our token-level BPD bounds are also predictive of downstream generalization and achieve 98.9% and 99.4% correlation with downstream perplexity and error, respectively.

In short, our bounds go significantly beyond the observation that the empirical distribution converges to the true distribution, and are predictive of generalization on downstream tasks. Achieving a non-vacuous token-level bound requires generalization.

### 4.4 Token-Level Prediction Smoothing

Rather than using a single label smoothing $\alpha$ for all data points, we propose to use the network itself to determine which tokens warrant more confidence and which ones require more smoothing to limit their worst-case behavior. We perform token-level prediction smoothing by adding a linear head to the LLM that outputs the probability $\alpha$ for each token, such that $p_h(x_i|x_{<i}) = \left(1 - \alpha_\theta(x_{<i})\right)p_\theta(x_i|x_{<i}) + \alpha_\theta(x_{<i})/V$. The training objective corresponds to the upper bound in Equation (2) rather than the empirical risk alone, where the $\alpha$ parameter factors into the bound via the interval size $\Delta_i = \log_2\left(1 + (1 - \alpha_\theta(x_{<i}))V/\alpha_\theta(x_{<i})\right)$. Therefore, the values of $\alpha_\theta(x_{<i})$ are adjusted to achieve the best trade-off between the empirical risk and the compressed model size. We perform this optimization post-training using a subset of the training dataset.

We demonstrate in Figure 4(Left) that using this token-dependent $\alpha$ significantly improves the value of the bounds. In Figure 4 (Middle), we compare to the setting where the optimal $\alpha$ is obtained through a grid search, and in Figure 4(Right) we examine the distribution of $\alpha$ produced by the model.

## 5 Compressing LLMs to Minimize Complexity

In shifting from document-level to token-level bounds, the number of data points $m$ increases considerably, and thus we can afford to pay significantly more bits in the complexity of the compressed model. In this new regime, the SubLoRA compression technique becomes very restrictive.

### 5.1 Efficient Nonlinear Parametrizations

In addition to LoRA, we explore two expressive nonlinear parametrizations $f(\theta)$ that make efficient use of the parameter space: Kronecker structures [15] and Monarch matrices [9]. We can use these nonlinear parametrizations directly, or in conjunction with subspace compression, parametrizing the full parameters as $\theta = \theta_0 + f(Pw)$ for a projection matrix $P \in \mathbb{R}^{D \times d}$. After training, the parameters are quantized as in and coded using arithmetic coding. We describe these structures below.

**LoRA.** With LoRA [19], the weight matrices of linear layers are parametrized via low rank updates. Each weight matrix $W \in \mathbb{R}^{a \times b}$ is parametrized $W = W_0 + AB$ for $A \in \mathbb{R}^{a \times r}, B \in \mathbb{R}^{r \times b}$ with a small rank $r$, where $W_0$ is given by the initialization and $A, B$ form the trainable parameters in each layer. Rather than considering only self-attention layer weights [19, 32], we extend SubLoRA to all linear layers in the model and compress the biases and layernorm weights in the subspace projection.

**Kronecker Product.** We can represent $W$ as a Kronecker product $W = A \otimes B$, where $\otimes$ is the Kronecker product, $A \in \mathbb{R}^{a_1 \times b_1}, B \in \mathbb{R}^{a_2 \times b_2}$ and $a_1 a_2 = a, b_1 b_2 = b$, which reduces the parameters over the dense layer. This approach has been used in recent work for parameter-efficient finetuning [15] and as an alternative structure for pretraining.

**Monarch Matrices.** We also consider Monarch matrices [9], which employ two block diagonal matrices $A$, and $B$ typically with $A$ and $B$ formed by $\sqrt{a}$ blocks of size $\sqrt{a} \times \sqrt{b}$ and a reshape or permutation operation $R$: $W = ARB$. The matrix multiplication is implemented by reshaping the input axis $a$ into $(\sqrt{a}, \sqrt{a})$, applying matrix $A$ as a batched matrix multiply on one axis, and then applying $B$ to the other axis by permuting the axes. Monarch matrices have shown considerable promise as an expressive and hardware-efficient replacement for linear layers.

### 5.2 QuIP 2-Bit Quantization of LLM

In addition to pretraining LLMs in efficient nonlinear subspaces, we explore recent post-training quantization methods to reduce the model complexity. Quantization with Incoherence Process (QuIP) compresses LLM weights to a smaller number of bits while preserving model performance [6].

**Adaptive Rounding.** For a weight matrix $W \in \mathbb{R}^{a \times b}$, QuIP minimizes the proxy quadratic objective $\ell(\hat{W}) = \mathbb{E}[\|(\hat{W} - W)x\|^2] = \text{tr}((\hat{W} - W)H(\hat{W} - W)^\top)$, where $\hat{W} \in \mathbb{R}^{a \times b}$ are the quantized

| Compression Approach | BPD Bound | Top-1 Error | Top-10 Error | Top-100 Error |
|---|---|---|---|---|
| SubLoRA [32] | 10.49 | 90.44 | 71.33 | 49.77 |
| Enhanced SubLoRA (Ours) | 10.44 | 89.38 | 69.54 | 49.84 |
| Enhanced LoRA (Ours) | 7.85 | 78.15 | 52.48 | 31.64 |
| Monarch Only (Ours) | **7.65** | **75.87** | **47.47** | **28.34** |
| Kronecker Only (Ours) | 8.03 | 80.80 | 52.77 | 30.14 |
| Kronecker + Subspace (Ours) | 10.02 | 88.75 | 67.91 | 47.14 |
| Random Guess | 15.62 | 99.99 | 99.98 | 99.80 |

Table 1: **Non-vacuous generalization bounds using different compression techniques for GPT2 pretraining.** We find that with the larger complexity budget afforded by the token-level bounds, subspace compression is no longer necessary or even beneficial for the bounds. Of the structures we consider, the Monarch parametrization performs best.

weights, $x \in \mathbb{R}^b$ is a vector drawn randomly from a calibration set, and $H$ is the second moment matrix of these vectors used as a proxy Hessian [6].

**Incoherence Processing.** Based on the observation that incoherences between the weights $W$ and the proxy Hessian $H$ benefit quantization, QuIP further applies incoherence post-processing using Kronecker products of random orthogonal matrices $U \in \mathbb{R}^{a \times a}, V \in \mathbb{R}^{b \times b}$ such that $\tilde{H} \leftarrow V H V^\top, \tilde{W} \leftarrow U W V^\top$. Here $U = U_1 \otimes \cdots \otimes U_k$ and $V = V_1 \otimes \cdots \otimes V_k$.

Subsequent work like QuIP# improves upon QuIP by using randomized Hadamard transform and vector quantizations [48]. To compute the compressed size $C(h)$ of QuIP-quantized models, we use `gzip` [12] to compress the quantized model checkpoint and obtain the term $C(h)$ as the bits required for the storage afterwards.

# 6  Non-Vacuous Bounds for LLMs with Billions of Parameters

We compute generalization bounds for: (i) models that are trained through non-linear subspace compression in the form of LoRA, Kronecker product or Monarch matrices on the OpenWebText dataset, then quantized using the same setup as Lotfi et al. [32], or (ii) models that are pretrained on a dataset other than the OpenWebText dataset – or on datasets that might have the OpenWebText as a subset– and made publicly available. For the pretrained models, we either apply aggressive quantization, which is the case for GPT2, or use QuIP 2-bit, 3-bit and 4-bit publicly-available quantized models, which is the case for LLaMA. In the pretrained LLMs setting, we evaluate our bounds for both the OpenWebText (9B tokens) and Amber (1.2T tokens) datasets. In both settings, we obtain highly compressed models that lead to non-vacuous generalization bounds. We also compute token-level generalization bounds for antibody design, a task where conditioning on contexts from the training dataset arises naturally. Finally, we investigate the effect of aggressive compression on memorization vs. reasoning in LLMs. We provide all the experimental details in Appendix B.

## 6.1  Token-level Bounds via Nonlinear Parametrizations

As discussed in Section 5.1, we experiment with LoRA in addition to the Kronecker and Monarch subspace parametrizations in order to train compressed versions of GPT2 small (124M parameters). Compared to previous work, we enhance both LoRA and SubLoRA by not only applying the low-rank decomposition to the attention layers and the linear head, but to all the fully-connected layers in the LLM. Additionally, we train all the bias and layer normalization parameters instead of keeping them fixed at their values at initialization. We also use rotary position embeddings [46] to directly encode the positional information into the LLM. Combined with our proposed token-level optimization of the label smoothing probability $\alpha$, we significantly improve upon the LoRA subspace compression, as shown in Table 1. It is worth noting the LoRA alone led to vacuous BPD document-level bounds in Lotfi et al. [32] while our version is non-vacuous.

Among all subspace compression strategies that we explore in Table 1, Monarch without subspace leads to the tightest token-level bound. In fact, the substantial scale of our dataset, comprising 9 billion tokens, significantly changes the trade-off between the empirical risk and the compressed model size compared to previous work, since the compressed size factor in the bound is divided by the

| Model | BPD | Top-1 Error (%) | Top-100 Error (%) |
|---|---|---|---|
| GPT2 (124M) | **7.61** | **74.82** | **26.98** |
| GPT2 (355M) | 8.50 | 79.19 | 32.72 |
| GPT2 (774M) | 10.47 | 89.50 | 44.23 |
| Random Guess | 15.62 | 99.99 | 99.80 |

Table 2: Pretrained GPT2 models achieve non-vacuous bounds for next token prediction on OpenWebText through post-training quantization only and without altering the pretraining.

| Model | BPD | Top-1 Error (%) | Top-100 Error (%) |
|---|---|---|---|
| LLaMA2-7B | **4.28** | **47.50** | **12.56** |
| LLaMA2-13B | 4.51 | 47.85 | 14.44 |
| LLaMA2-70B | 6.39 | 58.26 | 25.04 |
| Random Guess | 14.97 | 99.99 | 99.68 |

Table 3: Pretrained LLaMA2 models achieve non-vacuous token-level bounds for next token prediction on the Amber dataset via 2-bit post-training QuIP quantization only.

size of the dataset. Consequently, we have greater flexibility in selecting larger models that achieve an improved empirical risk. In this setting, the Monarch parametrization achieves the best trade-off between the empirical risk and the compressed size of the model as shown in Table 1, followed by LoRA and Kronecker. Monarch and Kronecker also perform best in terms of the validation loss, as shown in Figure 1(Right). The new trade-off between the empirical risk and the compressed size of the model also explains why subspace compression is no longer beneficial in obtaining tighter bounds compared to previous work, as further reducing the number of trainable parameters through linear subspace projection leads to a worse trade-off between the empirical performance of the compressed model and its compressed size.

## 6.2 Non-vacuous Bounds for Pretrained LLMs: GPT2, LLaMA1 and LLaMA2

Intensive quantization is another way we can achieve model compression, and therefore tighter generalization bounds. We explore the setting where we only apply post-training quantization to pretrained LLMs and compute the corresponding token-level generalization bounds.

**Pretrained GPT2 models.** We apply the post-training quantization [31] to the publicly available GPT2 models [39] of sizes 124M (GPT2 small), 354M (GPT2 medium), and 773M (GPT2 large) parameters that were pretrained on the WebText dataset and report the numbers in Table 2. We find that GPT2 small not only yields non-vacuous bounds, but these bounds are quite comparable to those obtained using aggressive compression techniques in Table 1. GPT2 medium and large also achieve non-vacuous bounds despite having almost a billion parameters.

**Pretrained LLaMA models.** In this set of experiments, we use pretrained and pre-quantized publicly available LLaMA1, LLaMA2 and LLaMA2-Chat models and plug in their empirical risk and compressed size directly into our token-level bounds. We report the bounds obtained for 2-bit LLaMA2 in Table 3. The full set of results is reported in Table 8. The bounds are computed for the next token prediction task on the Amber dataset, which contains 1.2T tokens. We obtain non-vacuous bounds for these models despite their large scale, ranging from 7 billion to 70 billions parameters. Our experiments show that the LLaMA2-Chat models achieve worse generalization bounds as reported in Table 8 and Figure 1(Left), demonstrating that fine-tuning Chat models for dialogue use cases hurts their generalization performance on next token prediction. Although we do not know what data was used to pretrain the LLaMA models, our bounds remain valid since they do not require for the models to be trained on the same data that the empirical risk is evaluated on.

**High-quality text generation.** A significant limitation of document-level bounds is that the SubLoRA model achieving the best document-level bound generates un-grammatical, low-quality text as demonstrated by Lotfi et al. [32] and shown in Table 9. In contrast, our top-performing model in terms of token-level BPD bounds on the OpenWebText dataset, which is the quantized GPT2 small model, generates high-quality text, ensuring a unique combination of practical usefulness and tight guarantees on the population risk.

## 6.3 Token-Level Generalization Bounds on Antibody Sequences

In addition to natural languages, our token-level generalization bounds are particularly descriptive of antibody design in biology. An antibody sequence is usually composed of 20 different amino acid tokens to bind to a target of interest. In therapeutic antibody design, biologists propose mutations to existing antibody sequences by changing the amino acid tokens at specific positions in the sequence.

Recent works have shown that LLMs pretrained on large antibody datasets can be used to propose mutations conditioned on starting antibody sequences [44, 2]. Our token-level generalization bounds match the settings by bounding the expected next amino acid token negative log likelihood averaged over training contexts that serve as starting sequences for iterative mutations. In Table 7, we show that language models based on the Mistral 7B architecture pretrained on a processed subset of the Observed Antibody Sequences (OAS) from scratch achieves non-vacuous token-level generalization bounds [20, 35, 2]. Details of these experiments can be found in Appendix B.9

### 6.4 Contextualizing GPT2 Bounds Against Markov Chains

The best token-level bound that we achieve for BPD on the OpenWeb-Text dataset is 7.6. But what does this value exactly mean? One might consider the possibility that our bounds are describing only the simplest components of fitting the data that exist in the model, such as the predictions of a 0th or 1st order Markov chain [34].

| Training Context Length | 0 | 1 | 2 | 4 | 1024 |
|---|---|---|---|---|---|
| GPT2-S-Quantized | 13.9 | 11.1 | 9.0 | 7.9 | **7.6** |
| Markov Chain | 11.3 | 10.5 | 15.3 | 22.4 | - |

Table 4: Our LLM bounds provide a much stronger statement than what would be explained by low order Markov models.

In Table 4, we show that this is not the case, by explicitly training a sparse $k$-th order Markov chain on OpenWebText and computing our token-level bounds for the result. Sweeping over different numbers of n-grams to use for the Markov chains, our bounds for these models cap out at $10.5$ BPD and rapidly degrade with higher order as more statistics need to be stored. We also train and compress versions of GPT2 that are restricted to only seeing $k$ tokens as context, mirroring the restrictions of the Markov chains. We find that for the simple 0 and 1st order Markov chains, our compression via the transformer is slightly worse. However, the LLM performs much better for higher orders.

### 6.5 Memorization vs. Reasoning

LLMs are capable of memorizing facts from their pretraining data, but they also can learn highly structured patterns. As we compress a model more and more, it must lose its ability to recall memorized facts, but it may still remember patterns, since they are compressible. In this section, we examine the difference between memorization and reasoning by measuring the ability of LLMs to compress structured and unstructured sequence data. To generate structured sequences, we first use short binary expression trees to generate numerical sequences of integers [17]. These sequences are highly compressible as they are generated using short and deterministic programs. To generate unstructured sequences, we collect the set of all unique integers from the structured sequences and form random sequences composed of IID samples from the set of unique integers (see Appendix B.6 for details). We train standard GPT2 models from scratch on structured and random sequences separately. In Figure 3, we show the integer prediction training accuracy with varying degrees of post-training quantization. We observe that as models are quantized more aggressively, i.e. the number of quantization levels decreases, they forget unstructured sequences far faster than structured sequences. These results parallel the findings of Jin et al. [21] who show that smaller models can retain in-context learning capabilities but lose their ability to recall facts.

## 7 Conclusion

In this work, we introduced novel token-level generalization bounds for LLMs which are able to accommodate the non-IID nature of the tokens within the training corpus. Combined with different compression techniques, we achieve non-vacuous generalization bounds for LLMs with up to 70 billion parameters. The compressed models for which we construct our bounds are capable of producing high quality text, unlike those in prior work. While there is still have a gap to close between the typical validation BPD and the constraint of our bounds, our bounds are predictive of generalization and provide insights into model behaviour.

In future work, one could envision constructing new bounds that make use of the independence structure between documents and then the non-independent structure within documents to achieve the best of both. It would also be exciting to further explore the development of these bounds for new downstream predictive tasks, in the vein of the antibody design task we briefly consider here.

## Acknowledgements

We thank Alan Amin for helpful discussions and anonymous reviewers for helpful feedback. This work at NYU is supported by NSF CAREER IIS-2145492, NSF CDS&E-MSS 2134216, NSF HDR-2118310, BigHat Biosciences, Capital One, and an Amazon Research Award.

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

# A Token-Level Martingale Bound

## A.1 Proof of the Main Theorem

**Theorem A.1.** *With probability at least $1 - \delta$ over the randomness in a sampled sequence $x_1, x_2, \ldots, x_m$, if the negative log likelihood of a model $h \in \mathcal{H}$ can be bounded $-\log_2 p_h(\cdot|x_{<i}) \in [a, a + \Delta_i]$ for some $\Delta_i$ (possibly a function of $h$), then the negative log likelihood of the data of a given hypothesis $h$ satisfies*

$$\frac{1}{m}\sum_{i=1}^{m}\mathbb{E}[-\log_2 p_h(X_i|x_{<i})|x_{<i}] \leq -\frac{1}{m}\log_2 p_h(x_{\leq m}) + \hat{\Delta}\sqrt{\frac{\log 1/P(h) + \log 1/\delta}{2m}}, \quad (4)$$

*where $\hat{\Delta} = \sqrt{\frac{1}{m}\sum_{i=1}^{m}\Delta_i^2}$, the expectation is taken over $X_i \sim p(X_i|x_{<i})$ from the data generating process, and $P(h)$ is any normalized prior over a discrete hypothesis space $\mathcal{H}$ that does not depend on $\{x_i\}_{i=1}^{m}$.*

*Proof sketch.* The proof of Theorem 4.1 is an application of Azuma's inequality [3] and can be broken down into the following steps:

- Construct a martingale difference sequence from the difference between the NLL on token $x_i$, and its expectation given the tokens $x_{<i}$. From the boundedness of NLL one can show that the differences are bounded.

- Apply Azuma's inequality for each hypothesis, choosing failure probability proportional to the chosen prior $P(h)$.

- Perform a union bound of the failure probabilities over all hypotheses. If all of the hypotheses satisfy the bound simultaneously, then so does the data dependent hypothesis $h^*$.

*Proof.* Given the autoregressive predictions $R(h, x_i, x_{<i}) := -\log_2 p_h(x_i|x_{<i})$ where $x_{<i} := \{x_1, x_2, \ldots, x_{i-1}\}$. Let $\{x_i\}$ denote the actual values of the sequence that were found empirically, and $\{X_i\}$ be the random variables for these quantities.

The collection of random variables (indexed by $i$) $Z_i = \mathbb{E}[R(h, X_i, x_{<i})|x_{<i}] - R(h, X_i, x_{<i})$ form a Martingale difference sequence with respect to $x_{<i}$. Note here that the expectation is over the distribution $X_i \sim p(X_i|x_{<i})$. From the construction, $\mathbb{E}[Z_i|x_{<i}] = 0$ and the sequence is bounded: $A_i = \mathbb{E}[R(h, X_i, x_{<i})|x_{<i}] - a \leq Z_i \leq \Delta_i + \mathbb{E}[R(h, X_i, x_{<i})|x_{<i}] - a = B_i$, with $B_i - A_i = \Delta_i$.

$\Delta_i$ may depend on $x_{>i}$ but only through it's dependence on the hypothesis $h(\{x\}_{i=1}^{m})$. For a fixed $h$ we may conclude that $\sum_{i=1}^{m} Z_i$ is bounded difference Martingale sequence (with respect to $\{x_{<i}\}_{i=1}^{m}$), and we can apply Azuma's inequality [3] to derive that for any $t > 0$:

$$P\left(\sum_{i=1}^{m} Z_i > mt\right) \leq \exp\left(-2m^2 t^2 / \sum_{i=1}^{m}\Delta_i^2\right)$$

$$P\left(\frac{1}{m}\sum_{i=1}^{m} Z_i > t\right) \leq \exp\left(-2mt^2/\hat{\Delta}^2\right).$$

Judiciously choosing

$$t(h) = \hat{\Delta}\sqrt{\frac{\log 1/P(h) + \log 1/\delta}{2m}},$$

we have that $P\left(\frac{1}{m}\sum_{i=1}^{m} Z_i > t(h)\right) = P(h)\delta$.

Applying a union over the events $\bigcup_{h \in \mathcal{H}}\left[\frac{1}{m}\sum_{i=1}^{m} Z_i(h) > t(h)\right]$, we have

$$P\left(\frac{1}{m}\sum_{i=1}^{m} Z_i > t(h)\right) \leq \sum_{h} P(h)\delta = \delta,$$

therefore $P\left(\frac{1}{m}\sum_{i=1}^{m}Z_i \leq t(h)\right) > 1 - \delta$. Unpacking the definition of $Z_i$, we have that with probability at least $1 - \delta$

$$\frac{1}{m}\sum_{i=1}^{m}\mathbb{E}[R(h, X_i, x_{<i})|x_{<i}] \leq \frac{1}{m}\sum_{i=1}^{m}R(h, x_i, x_{<i}) + \hat{\Delta}\sqrt{\frac{\log 1/P(h) + \log 1/\delta}{2m}}.$$

Expressed in terms of the log likelihood, we can write this as:

$$\frac{1}{m}\sum_{i=1}^{m}\mathbb{E}[-\log_2 p_h(X_i|x_{<i})|x_{<i}] \leq -\frac{1}{m}\log_2 p_h(x_{\leq m}) + \hat{\Delta}\sqrt{\frac{\log 1/P(h) + \log 1/\delta}{2m}}$$

$\square$

## A.2 Empirical Risk Subsampling

We evaluate our bounds for the OpenWebText and Amber datasets which contain 9 billion and 1.2 trillion tokens, respectively. Computing the exact empirical risk for these datasets would be prohibitively expensive. Therefore, we use subsampling for the evaluation of the empirical risk to accelerate bound computation. In Equation (2), we use the following inequality which holds with probability at least $1 - \delta_2$:

$$-\frac{1}{m}\log_2 p_h(x_{\leq m}) \leq -\frac{1}{n}\sum_{j=1}^{n}\log_2 p_h(x_{\sigma(j)}|x_{<\sigma(j)}) + \hat{\Delta}\sqrt{\frac{\log 1/\delta_2}{2n}} \qquad (5)$$

for a subsample of size $n$ where $\sigma$ is a random permutation. We choose $\delta_1$ in Equation (2) with respect to a new overall failure probability $\delta$ to be $\delta_1 = \delta n/(n + m)$ and choose $\delta_2 = \delta m/(n + m)$ so that the overall failure probability is still $\delta$. The proof is simple and similar to that provided in Lotfi et al. [32].

# B    Experimental Details

## B.1    Pretraining with Nonlinear Parametrizations

To achieve the necessary model compression level for computing non-vacuous bounds, we pretrain GPT2 Small with 124 million parameters on the OpenWebText[3] dataset based on the nanoGPT implementation[4] [39]. We parametrize the linear layers of `CausalSelfAttention`, `MLP`, and the `LinearHead` of the GPT2 models with our nonlinear compression techniques (LoRA, Kronecker, Monarch), where we use a bias vector except for the `LinearHead` layer. For LoRA and Kronecker, we use weight tying between the token embedding and the final `LinearHead` layer parameterized by nonlinear compression techniques. We also train the layer norm parameters in addition to all of the nonlinear projection parameters applied to the linear layers. For Monarch, we only train the linear layers parameterized by Monarch matrices. We also combine the three nonlinear parametrizations with linear subspace projection, where all the trainable parameters $\theta$ are projected into a subspace of parameters $w$ using a projection matrix $P$, such that $\theta = \theta_0 + Pw$. We vary the dimension of $w$ as a hyperparameter in the bound evaluation.

For all the pretraining experiments, we use a batch size of $8$, a sequence length of $1024$, and a standard AdamW optimizer [30] with a learning rate of $0.0002$. We perform a learning rate warm-up for $500$ iterations, and we apply rotary embedding [46] to all three nonlinear parametrizations.

### B.1.1    Hyperparameter Sweeps for LoRA

**LoRA.** We sweep over LoRA rank values $r \in \{1, 4, 16, 32, 64, 128, 256\}$. We choose a learning rate of $0.0002$ with a LoRA dropout value of $0.1$ and LoRA alpha value of $32$.

**SubLoRA.** We report the rank $r$ and the corresponding subspace dimension values that we sweep over for SubLoRA in Table 5.

---

[3] `http://Skylion007.github.io/OpenWebTextCorpus`
[4] `https://github.com/karpathy/nanoGPT`

| Rank $r$ | Subspace Dimension $d$ |
|:---:|:---:|
| 1 | 25000 |
| 4 | 50000 |
| 8 | 50000 |
| 16 | 50000 |
| 32 | 10000, 750000 |
| 64 | 25000, 2000000 |
| 128 | 7000000, 15000000 |

Table 5: Hyperparameter sweep for SubLoRA. For all the SubLoRA pretraining experiments, we use a learning rate of $0.0002$, a LoRA dropout value of $0.1$, and a LoRA alpha value of $32$.

### B.1.2 Hyperparameter Sweeps for Kronecker

For the Kronecker factorization $W = A \otimes B$, we choose the matrices $A$ and $B$ such that $A \in \mathbb{R}^{a_1 \times b_1}, B \in \mathbb{R}^{a_2 \times b_2}$ where $a_1 a_2 = a$ and $b_1 b_2 = b$. We sweep over all possible combinations of $\{a_1, a_2\}$ and $\{b_1, b_2\}$ by performing prime factorizations with multiplicity on the numbers $a, b$ and enumerating all possible combinations. All of our Kronecker pretraining experiments use a learning rate of $0.0002$.

### B.1.3 Hyperparameter Sweeps for Monarch

For the Monarch parametrization, we relax the restriction for the number of blocks to be strictly $\sqrt{a}$ and instead by a number divisible by $a$ to sweep over different numbers of blocks. We also perform experiments for Monarch where we are using absolute position encodings and experiments where we are only applying the Monarch factorization to the attention layers and the linear classification heads.

## B.2 Quantization

**Quantization.** Following Lotfi et al. [31], we apply post-training quantization of the trainable weights that correspond to the subspace parameters and/or the LoRA, Kronecker, Monarch parameters along with layer norm weights depending on the compression setup. In this case, we map the pretrained weights into a significantly smaller number of quantization clusters. The quantized vector $\hat{w} = [\hat{w}_1, \ldots, \hat{w}_d]$ can be constructed from the original weights vector $w = [w_1, \ldots, w_d]$ by assigning these weights to different clusters $c = [c_1, \ldots c_L]$, where $\hat{w}_i = c_q$ such that $q = \operatorname{argmin}_k |w_i - c_k|$. The quantization clusters $c$ are learned alongside $w$, such that we optimize the empirical risk and the compressed size of the model as well.

**Experiments on QuIP-quantized Models.** We compute token-level bounds on pretrained LLaMA1 and LLaMA2 models [47] quantized with QuIP with publicly-available checkpoints [42]. Although we do not know what data was used to pretrain these models, we can evaluate the generalization bound on the Amber dataset and consider other tokens used in training as a data-dependent prior.

## B.3 Bounds Evaluation

In the sequence of text, we use end of text tokens (EOT) which separate the documents. In this way, we can consider concatenating many documents together to form one long sequence. As a result of the EOT tokens and the structure of the text, the distribution $p(x_i|x_{<i})$ can be simplified into $p(x_i|x_k, x_{k+1}, \ldots x_{i-1})$ where $k$ is the index of the most recent EOT token because the documents are sampled independently. In the evaluation of the LLM we likewise have no dependence on tokens outside the given document in question.

To compute token-level bounds, we evaluate all of our generalization bounds with failure probability $\delta = 0.05$ and subsample size of $n = 10,0000$ tokens from the OpenWebText training dataset of size $m = 9$ billion tokens or the Amber dataset of size $m = 1.2$ trillion tokens.

**Evaluation metrics.** In addition to reporting generalization bounds for the bits-per-dimension (BPD) loss, we also report the bounds that we obtain for the Top-1, Top-10 and Top-100 error. The Top-$k$ error refers to the 0-1 error in predicting the next token among the top-$k$ predictions of the

model. For instance, the Top-1 error for token $x_i$ is defined as $\mathbf{1}[\text{argmax}_{x_j} \, p(x_j | x_{<i} = x_{<i}) = x_i]$, where $\text{argmax}$ operates over tokens $x_j$ across the vocabulary. We extend this definition to the Top-k error and define it as $\mathbf{1}[x_i \in \text{argmax}_{x_j,k} \, p(x_j | x_{<i} = x_{<i})]$, where the $\text{argmax}$ operator here selects the top-$k$ tokens predicted by the model according to its next token probability distribution $p(x_j | x_{<i} = x_{<i})$. Our bound in Equation (2) applies not only to the log likelihood but to any bounded risk, and therefore can be computed for the Top-$k$ error since it is bounded between $0$ and $1$. We call a Top-$k$ error bound *vacuous* when the bound is larger than the random guess top-$k$ error equal to $1 - k/V$, where $V$ is the vocabulary size.

## B.4 Correlation with Downstream Performance

We retrieve the downstream task performance of difference GPT2 variants ranging in scale from 117M to 1.5B averaged over the downstream datasets as shown in Table 6. To obtain an approximation of the conditional BPD expectation that we bound in Equation (2), we resample $x_i$ from a LLaMA2-7B given fixed training contexts $x_{<i}$ from the Amber dataset. We use a sample size equal to $10,000$ samples.

| Model Size | LAMBADA (PPL) | LAMBADA (ACC) | CBT-CN (ACC) | CBT-NE (ACC) | WikiText2 (PPL) | PTB (PPL) | WikiText103 (PPL) | 1BW (PPL) |
|---|---|---|---|---|---|---|---|---|
| 117M | 35.13 | 45.99 | 87.65 | 83.4 | 29.41 | 65.85 | 37.50 | 75.20 |
| 345M | 15.60 | 55.48 | 92.35 | 87.1 | 22.76 | 47.33 | 26.37 | 55.72 |
| 762M | 10.87 | 60.12 | 93.45 | 88.0 | 19.93 | 40.31 | 22.05 | 44.575 |
| 1542M | 8.63 | 63.24 | 93.30 | 89.05 | 18.34 | 35.76 | 17.48 | 42.16 |

Table 6: Zero-shot downstream task performance for GPT2 models with different model sizes as reported in Radford et al. [39].

## B.5 Markov Chain Comparison

For training the Markov chains, we reuse the Byte Pair Encoding (BPE) tokenization to separate out the effect of the tokenizer. We apply prediction smoothing at level $\alpha = 0.1$ to the Markov models to give them nonzero probability to ngrams that have not been seen in the training data and limit the worst case NLL of a single token.

For constructing generalization bounds with the Markov chain, we upper bound the complexity term $\log 1/P(h)$ similarly to the large language models by performing quantization and compression. We store and update the Markov chains sparsely, which becomes necessary when considering the high order variants. In storing the model, we use a dictionary mapping each prefix concatenated with the following token to a count. The counts can then be converted into probabilities by normalizing by the count containing just the prefix. We quantize the counts and store them in 16 bits, and we store the keys using a basic encoding. For training, we train on a subsample of $10^6$ tokens from the training corpus, sufficient for the performance of the Markov chains to converge.

## B.6 Memorization Experiment

Following Goldblum et al. [17], we select a complexity value of $4$, which reflects the difficulty of the task, and a sequence length of $30$ and generate $984$ sequences as the training dataset for structured sequences. To build our baseline random sequences, we collect all unique integers in the generated sequences into a set. We then sample integers IID from a uniform distribution over the set of unique integers from the structured sequences to build the baseline dataset. Our vocabulary size is $12$ as we only have integers, the beginning of text token, and an additional delimiter token. The delimiter tokens are placed between distinct numbers during our tokenization process. We use a GPT-2 Small model with 124M parameters and train it on the structured and random sequences separately with a learning rate of $0.0001$ for 1000 epochs. Our quantization procedure is the same as described in Appendix B.2. We show the results for this experiment in Figure 3.

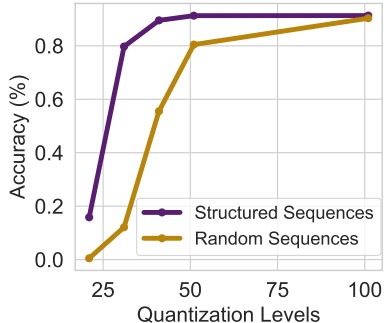

Figure 3: **As language models are compressed, they retain their understanding of patterns, but they forget highly random and unstructured data rapidly.** Experiments performed on GPT-2 models with datasets created as detailed in Section 6.5. Compression performed via post-training quantization where lower quantization levels reflect more aggressive compression..

### B.7 Amber Dataset

We use a subset of the pretraining dataset for Amber 7B LLM [29] for our bound evaluations. This dataset contains RedPajama V1 [8] (arxiv, C4, GitHub, StackExchange, Wikipedia), StarCoder [26] (The Stack), RefinedWeb [37] (CommonCrawl) with around 1.2 trillion tokens. We tokenize the entire dataset using a LLaMA tokenizer and then sample tokens from a uniform distribution over the tokenized dataset.

### B.8 Compute Budget

For all our pretraining experiments with the three proposed compression techniques, we run each experiment for 5 days on 4 GPUs in parallel that are of type A100sor RTX8000. For the bound computation experiments, we use a single GPU of any type and a subsample size of $10,000$ samples. The running time varies between $1$ to $8$ hours depending on the model and the dataset. All other experiments are performed on a single GPU of any type.

### B.9 Bounds on Antibody Sequences

#### B.9.1 Datasets

An antibody consists of both the light chain and the heavy chain amino acid sequences. Among these sequences, there are collections of sequences called the clonal family that our immune systems developed to bind to targets. For our experiments, we select all human heavy chain amino acid sequences from the Observed Antibody Space (OAS) and keep all the clonal families with at least 25 sequences using the FastBCR filtering technique following [35, 49, 2]. The processed dataset contains around 908 thousand heavy chain clonal families. A single example in our dataset is thus a clonal family looking like [sequence 1, sequence 2, ..., sequence N] where $N \geq 25$.

There are in total 29 different tokens with 20 of them corresponding to 20 different amino acids. Let $[ClSep]$ be a separator token between sequences in a clonal family. We process our input example by forming the string "sequence 1 $[ClSep]$ sequence 2 $[ClSep]$ ... $[ClSep]$ sequence N" following [2]. This input example is tokenized and given to a language model using the next token prediction training objective.

#### B.9.2 Language Models

We use language model architectures that are based on the Mistral 7B architecture [20]. We scale down the Mistral architecture using 24 layers with a varying hidden state size of $(1024, 768, 512)$, resulting in our Mistral 377M, Mistral 212M, and Mistral 94M models, respectively following [2]. With a vocabulary size of 29 and a maximum context length of 2048, we train each of our models using 4 NVIDIA A100s for 48 hours and perform post-training quantization following Lotfi et al. [31].

| Compression Approach | Bits Per Dimension | Top-1 Error | Top-10 Error | Validation Loss |
|---|---|---|---|---|
| Mistral 377M | 2.41 | 31.60 | 26.46 | **0.28** |
| Mistral 212M | 2.06 | 26.25 | 21.07 | 0.30 |
| Mistral 94M | **1.62** | **19.40** | **14.59** | 0.30 |
| Random Guess | 4.86 | 96.56 | 65.51 | 1.46 |

Table 7: **Models pretrained on antibody sequences achieve non-vacuous token-level generalization bounds.** Language models pretrained on antibody sequences achieve non-vacuous bounds for next token prediction on a processed subset of Observed Antibody Sequences (OAS) through post-training quantization only. The vocabulary size of an antibody LLM is 29.

## C    Additional Results

### C.1    Token-Level Prediction Smoothing Optimization

In Figure 4, we show the bounds we obtain with and without optimizing the prediction smoothing probability for different numbers of trainable parameters. We observe that post-training optimization of $\alpha$ at the token-level yields significantly better bounds.

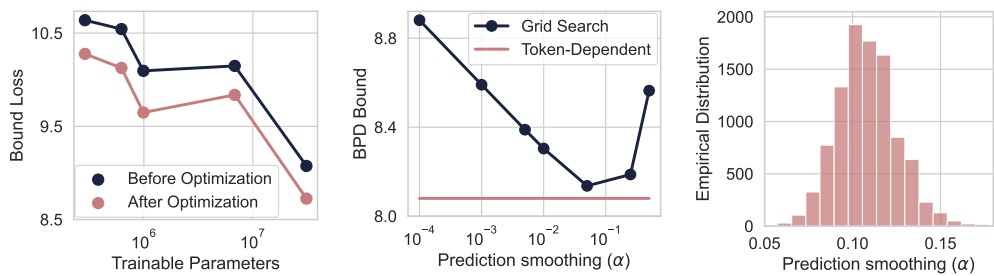

Figure 4: **Token-level prediction smoothing improves our bounds. Left:** After training, we optimize a conservative upper bound on the generalization bound that we would get from Equation (2) with respect to the $\alpha$ head parameters. Doing so yields a noticeable reduction in the value of the bound. **Middle:** BPD generalization bound as a function of a single global parameter chosen from a discrete number of values vs. the generalization bound for the token-dependent $\alpha$ after optimization. **Right:** Histogram of the values taken by $\alpha(x_{<i})$ over different inputs.

### C.2    LLaMA Bounds on Amber

In Table 8, we have the complete bounds computation results for LLaMA1, LLaMA2, LLaMA2 Chat with 2-bit, 3-bit, and 4 bit-quantization on the Amber dataset. The best bound is achieved by a LLaMA2 model with 2-bit quantization.

### C.3    Token-Level Bounds Are Predictive of Downstream Performance

We compute the direct correlation between our bounds and the downstream performance on the tasks reported in Table 6. In Figure 5 (Left $y$-axis), we plot the average zero-shot error (Error), defined as $1$ - the accuracy, and the perplexity (PPL) achieved by GPT2 small, medium and large on the downstream tasks in Table 6. On the right $y$-axis, we plot the token-level bounds achieved by the GPT2 models with different sizes on the OpenWebText dataset that they were partially trained on. Our token-level BPD bounds achieve $98.9\%$ and $99.4\%$ correlation with the downstream perplexity and error, respectively, and are indeed predictive of generalization on downstream tasks.

### C.4    Generated Text

In Table 9, we show the generated text by the model achieving the best bounds: the quantized GPT2 model that achieves the best token-level bounds on the OpenWebText dataset in our work, and the GPT2 model trained with SubLoRA that achieves the best document-level bounds in Lotfi et al. [32].

| Model | Bits per Dimension | Top-1 Error (%) | Top-10 Error (%) | Top-100 Error (%) |
|---|---|---|---|---|
| 2 bits | | | | |
| LLaMA1-7B | 4.29 | 48.08 | 22.82 | 12.83 |
| LLaMA1-13B | 4.60 | 48.87 | 24.23 | 14.59 |
| LLaMA1-30B | 5.37 | 52.91 | 28.06 | 19.14 |
| LLaMA1-65B | 6.10 | 56.63 | 32.29 | 24.14 |
| LLaMA2-7B | **4.28** | 47.55 | **22.48** | **12.56** |
| LLaMA2-Chat-7B | 4.54 | 49.10 | 24.18 | 13.50 |
| LLaMA2-13B | 4.52 | 47.85 | 23.54 | 14.44 |
| LLaMA2-Chat-13B | 4.77 | 49.82 | 24.95 | 15.10 |
| LLaMA2-70B | 6.14 | 56.24 | 32.61 | 24.32 |
| LLaMA2-Chat-70B | 6.40 | 58.26 | 34.16 | 25.04 |
| 3 bits | | | | |
| LLaMA1-7B | 4.37 | 47.42 | 22.87 | 13.63 |
| LLaMA1-13B | 4.80 | 48.97 | 25.23 | 16.14 |
| LLaMA1-30B | 5.70 | 53.54 | 29.91 | 21.63 |
| LLaMA1-65B | 6.73 | 59.56 | 36.14 | 28.08 |
| LLaMA2-7B | 4.35 | **47.15** | 22.75 | 13.62 |
| LLaMA2-Chat-7B | 4.65 | 48.84 | 24.23 | 14.24 |
| LLaMA2-13B | 4.76 | 48.45 | 24.67 | 15.95 |
| LLaMA2-Chat-13B | 5.06 | 50.90 | 26.26 | 16.66 |
| LLaMA2-70B | 6.77 | 59.35 | 36.27 | 28.56 |
| LLaMA2-Chat-70B | 7.08 | 61.66 | 38.00 | 29.30 |
| 4 bits | | | | |
| LLaMA1-7B | 4.50 | 47.52 | 23.53 | 14.52 |
| LLaMA1-13B | 5.02 | 49.96 | 26.46 | 17.47 |
| LLaMA1-30B | 6.05 | 55.55 | 32.09 | 23.93 |
| LLaMA1-65B | 7.27 | 62.56 | 39.38 | 31.54 |
| LLaMA2-7B | 4.49 | 47.64 | 23.64 | 14.53 |
| LLaMA2-Chat-7B | 4.83 | 49.49 | 25.15 | 15.12 |
| LLaMA2-13B | 4.96 | 49.46 | 25.67 | 17.21 |
| LLaMA2-Chat-13B | 5.27 | 51.61 | 27.23 | 18.12 |
| LLaMA2-70B | 7.33 | 62.53 | 39.89 | 32.11 |
| LLaMA2-Chat-70B | 7.68 | 65.32 | 41.59 | 32.87 |
| Random Guess | 14.97 | 99.99 | 99.96 | 99.68 |

Table 8: **Non-vacuous token-level generalization bounds for open-source pretrained LLM checkpoints on the Amber dataset.** All of these models were quantized post-training using QuIP# to different numbers of bits as shown above. All the bounds are non-vacuous compared to random guess performance.

The text generated by the model achieving the best generalization bounds in our work is visibly more coherent and grammatically correct. By switching from document-level to token-level bounds, obtaining non-vacuous bounds requires less restrictive compression techniques and therefore can be achieved for highly performant models that generate high-quality text and can be deployed in practice.

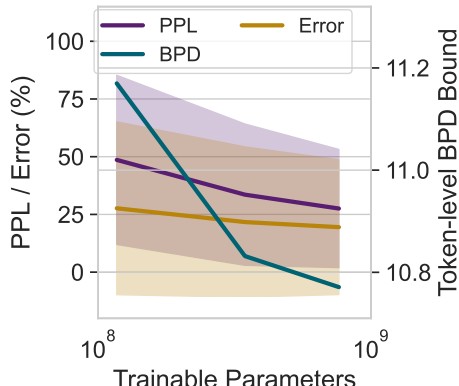

Figure 5: **Our bounds are predictive of generalization on downstream tasks.** On the left $y$-axis, we plot the average zero-shot error (Error) and the perplexity (PPL) achieved by GPT2 small, medium and large models pretrained with SubLoRA on downstream tasks reported in Table 6 of the original manuscript. On the right $y$-axis, we plot the the bounds achieved by GPT2 models on OpenWebText. Our bounds achieve **98.9**% and **99.4**% correlation with the perplexity and error, respectively.

| | Generated Text |
|---|---|
| GPT2 (124M) Quantized (BPD Bound: 7.61) | The study, published in Proceedings of the National Academy of Sciences, examined the relationships between brain activity, gene expression and inflammation in diseases including Alzheimer's disease, dementia, Parkinson's disease, glioblastoma and Alzheimer's disease. "Our study demonstrates that omega-3 fatty acids play a role in the link between inflammation and brain function," said lead author Dr Richard Collins, PhD, of Duke University's Duke Center for Bioethomics and Bioengineering. After controlling for. |
| GPT2 (124M) SubLoRA [32] | th he the startedt at its,, the a more be power and- by. S and, of of -'s on. The UK I The, are the on the the under, but the then the day,. The. The. It for the! a,. M an they first the the speak have times. cover that ( illegal In the day where I The who when and $ In We ̈:[{ ̈ As she I WeP spirituality. The all And one which a more says thought the other (ed 15: And P It as/ T - 2 But We The The theah It who the full of that to was 'The they (It As We A and each (. The It - We The M I" |

Table 9: **The best non-vacuous token-level bounds correspond to models that generate high quality text.** Examples of generated text from the GPT2 small quantized model that achieves the best token-level bounds compared to the SubLoRA-pretrained GPT2 small model in Lotfi et al. [32]. In contrast to the text generated by the best performing model in terms of BPD bounds by Lotfi et al. [32], our quantized GPT2 small generates significantly higher-quality text while simultaneously achieving the best BPD and Top-1/10/100 error bounds.

