# OpenReview forum: "Unlocking Tokens as Data Points for Generalization Bounds on Larger Language Models"
_NeurIPS.cc/2024/Conference — NeurIPS 2024 spotlight_

### Official Review · Reviewer_4zMP · 2024-07-06

**Soundness:** 3
**Presentation:** 3
**Contribution:** 3
**Rating:** 7
**Confidence:** 2

**Summary:**

The paper proposes token-level generalization bounds for large language models (LLMs), such as LLaMA2-70B, using less restrictive compression techniques like Monarch matrices, Kronecker factorizations, and post-training quantization. The authors argue that traditional document-level bounds are vacuous at this scale and introduce a method leveraging martingales for deriving tighter bounds, which not only hold theoretically but are also demonstrated through empirical validation.

**Strengths:**

1. **Originality**: The paper introduces a novel approach to computing generalization bounds at the token level, which is a significant departure from the document-level bounds prevalent in prior works.
2. **Technical Soundness**: The use of martingales and non-restrictive compression methods to derive generalization bounds is both innovative and robust, providing a solid theoretical framework backed by empirical results.
3. **Significance**: The ability to provide non-vacuous generalization bounds for LLMs as large as 70 billion parameters is highly significant, as it pushes the boundary of what is understood about LLM generalization in practical settings.
4. **Clarity**: The paper is well-written, with clear explanations of the methods and their implications, making it accessible to readers who may not be experts in the specific sub-field of machine learning.

**Weaknesses:**

No major weaknesses

**Questions:**

Can the generalization bounds proposed be integrated into the training regimen to enhance model generalization directly?

**Limitations:**

The paper lacks intuitive explanations for the proposed bounds, which might hinder understanding for readers not familiar with advanced statistical concepts in machine learning.

---

> ### Author Rebuttal · Authors · 2024-08-07
>
> Many thanks for your encouraging and thoughtful feedback! We respond to your questions below.
>
> **An intuitive application of our bounds to downstream tasks:**
>
> Inspired by your comments, we provide another intuitive application of our bounds. In this case, to a downstream scientific task. Our token-level generalization bounds are particularly descriptive of antibody design in biology. An antibody sequence is usually composed of 20 different amino acid tokens to bind to a target of interest. In therapeutic antibody design, biologists propose mutations to existing antibody sequences by changing the amino acid tokens at specific positions in the sequence. Recent works have shown that LLMs pretrained on large antibody datasets can be used to propose mutations conditioned on starting antibody sequences. **Our token-level generalization bounds match the antibody design setting by bounding the expected next amino acid token negative log likelihood averaged over training contexts that serve as starting sequences for iterative mutations.** In the table below, we show that language models based on the Mistral 7B architecture pretrained on a processed subset of the Observed Antibody Sequences (OAS) from scratch achieves non-vacuous token-level generalization bounds:
>
> |    Compression Approach    |    BPD Bound   |   Top-1 Error Bound   |   Validation Loss  |
> |               ---                           |          ---            |           ---                     |             ---             |
> |         Mistral 377M                |       2.41            |           31.60               |             0.28          |
> |         Mistral 212M                |        2.06           |           26.25               |             0.30          |
> |         Mistral 94M                  |        **1.62**     |           **19.40**          |             0.30         |
> |         Random Guess            |        4.86          |            96.56               |             1.46        |
>
> **Using generalization bounds for model optimization:**
>
> It is indeed possible to integrate the elements of our bounds directly into the training regimen to enhance model generalization directly. One example is quantization-aware training, which we in fact already use for post-training quantization of all pretrained models except the LLaMA models. In this scenario, we wish to map the pretrained weights of the neural networks into a significantly smaller number of quantization clusters. The quantized vector $\hat{w} = [\hat{w}_1,\dots,\hat{w}_d]$ can be constructed from the original weights vector $w = [w_1,\dots,w_d]$ by assigning these weights to different clusters $c = [c_1,\dots c_L]$, where $\hat{w}_i =c_q$ such that ${q= \operatorname{argmin}_k |w_i-c_k|}$. The quantization clusters $c$ are learned alongside $w$, such that we optimize the empirical risk and the compressed size of the model as well. We will add more details about our quantization-aware training procedure in the appendix. The focus of our work however is to compute non-vacuous generalization bounds that we can use to understand when and why LLMs generalize and provide a prescription for how to design better models in practice.
>
> We provide additional experiments in the general response showing that not only is the quantity we bound predictive of downstream performance, but the bounds themselves positively correlate with downstream performance. We also demonstrate that the trade-off between the empirical risk and the compressed size of the model highly depends on the compression scheme and the size of the training data. The additional figures can be found in the Rebuttal PDF.
>
> Inspired by your feedback, we also made sure to include additional intuitive explanations and details throughout the revised manuscript to make sure that the paper's content is accessible to readers.
>
> Thank you again for your supportive and positive review. We made a significant effort to revise our paper and run additional experiments in light of your feedback. We would appreciate it if you would consider raising your score in light of our response. Please let us know if you have any additional questions or comments.

---

> > ### Comment · Reviewer_4zMP · 2024-08-12
> >
> > I appreciate the authors response, which addressed all my questions. I increase my score to 7.

---

### Official Review · Reviewer_mAFv · 2024-07-09

**Soundness:** 3
**Presentation:** 4
**Contribution:** 3
**Rating:** 6
**Confidence:** 4

**Summary:**

This paper develops nonvacuous generalization bounds for modern language models. Specifically, this paper proves a token-level generalization bound, and applies different techniques (LoRA, 2 Kronecker Product, Monarch Matrices, and post-training quantization) to control the capacity of model class.

**Strengths:**

This paper is well-written: Theorem 3.1 is clean, and many compression techniques (parameter-efficient tuning, post-training quantization,etc.) and modern language models (GPT, LLAMA, etc.) are analyzed. I believe these are nice contributions.

**Weaknesses:**

The main weakness in my opinion is the left-hand side of eq. (2), since it uses contexts from the training data. I agree it is still a meaningful result, but it is also a little hard to interpret. Line 190 claims that "This figure confirms our intuition that the next token distribution is particularly diffuse at the beginning of a sentence, while it decreases for later tokens but remains relatively high. Given how diffuse the distribution is and the large number of possible sentences, it is broadly infeasible to make predictions on new resampled tokens from the empirical distribution alone." I am not fully convinced by this, since we can add some fixed prompt and only measure generalization error for the generated part.

**Questions:**

Figure 2 (right) claims that the left-hand side of eq. (2) is correlated with downstream performance. Specifically, the left y-axis plots accuracies of GPT-2 models, while the y-axis uses samples from the LLAMA model; if I understand correctly, the LLAMA model is treated as an oracle model here, since we do not know the true data-generation process. However, I feel this is a little indirect; can you instead compare downstream performance with your generalization bound (i.e., right-hand side of eq. (2)) for the same model? If this also works, then we can claim that not only the quantity we try to bound is meaningful, but the bound itself is also meaningful.

---

> ### Author Rebuttal · Authors · 2024-08-07
>
> We really value your thoughtful and supportive feedback! We provide several additional results inspired by your comments.
>
> **An intuitive application of our bounds to downstream tasks:**
>
> Inspired by your comments, we provide another intuitive application of our bounds. In this case, to a downstream scientific task. Our token-level generalization bounds are particularly descriptive of antibody design in biology. An antibody sequence is usually composed of 20 different amino acid tokens to bind to a target of interest. In therapeutic antibody design, biologists propose mutations to existing antibody sequences by changing the amino acid tokens at specific positions in the sequence. Recent works have shown that LLMs pretrained on large antibody datasets can be used to propose mutations conditioned on starting antibody sequences. **Our token-level generalization bounds match the antibody design setting by bounding the expected next amino acid token negative log likelihood averaged over training contexts that serve as starting sequences for iterative mutations.** In the table below, we show that language models based on the Mistral 7B architecture pretrained on a processed subset of the Observed Antibody Sequences (OAS) from scratch achieves non-vacuous token-level generalization bounds:
>
> |    Compression Approach    |    BPD Bound   |   Top-1 Error Bound   |   Validation Loss  |
> |               ---                           |          ---            |           ---                     |             ---             |
> |         Mistral 377M                |       2.41            |           31.60               |             0.28          |
> |         Mistral 212M                |        2.06           |           26.25               |             0.30          |
> |         Mistral 94M                  |        **1.62**     |           **19.40**          |             0.30         |
> |         Random Guess            |        4.86          |            96.56               |             1.46        |
>
>
> **Correlation between our bounds and the performance on downstream tasks:**
>
> Following your suggestion, we compute the direct correlation between our bounds and downstream performance on the tasks reported in Table 6 of the paper. In Figure 1(left y-axis) reported in the attached Rebuttal PDF, we plot the average zero-shot error (Error), defined as 1 - the accuracy, and the perplexity (PPL) achieved by GPT2 small, medium and large on downstream tasks, as reported in Radford et al. [1]. On the right y-axis, we plot the token-level bounds achieved by the GPT2 models with different sizes on the OpenWebText dataset that they were partially trained on. Our token-level BPD bounds achieve **98.9%** and **99.4%** correlation with the downstream perplexity and error, respectively, and are indeed predictive of generalization on downstream tasks. Given the significance of these results, we included them in our revised manuscript.
>
> **The distribution over next tokens being diffuse:**
>
> As you mentioned and as shown in Figure 2 (middle), the average entropy of the next token distribution conditioned on fixed contexts decreases for later token positions compared to the beginning of text token. Therefore, if we have a fixed prompt, this distribution might be less diffuse for the first token that we predict given the prompt. However, the main takeaway from this experiment actually pertains: even for later token positions, the distribution does not collapse entirely and the entropy is non-zero. For instance, if the prompt is composed of 127 tokens, then the average entropy for the next token is equal to 3 bits after transformation to the logarithm base 2, which corresponds to 8 choices for the next tokens. The number of choices is far greater than a single predetermined token, which implies that it is infeasible to make predictions on new resampled tokens from the empirical distribution alone. Therefore, non-vacuous token-level bounds are indicative of generalization beyond the training data.
>
> Thank you again for your detailed review. We put a significant effort into our response and would appreciate it if you could consider raising your score.
>
> ______
> References:
>
> [1] A. Radford, J. Wu, R. Child, D. Luan, D. Amodei, I. Sutskever, et al. Language models are 454 unsupervised multitask learners. OpenAI blog, 1(8):9, 2019.

---

### Official Review · Reviewer_iy2E · 2024-07-18

**Soundness:** 3
**Presentation:** 3
**Contribution:** 3
**Rating:** 6
**Confidence:** 3

**Summary:**

This paper presents a novel approach that computes non-vacuous compression-based generalization bounds for LLMs at the billion-parameter scale. Prior works could only achieve vacuous bounds for these large-scale models and rely on the assumption of IID documents. By leveraging the vast number of tokens in LLM training sets and properties of martingales, the authors derive non-vacuous bounds for LLMs that generate high-quality texts. Further, they showcase the tightness of the bounds by examining compression schemes including Monarch matrices and Kronecker factorizations, and post-training quantization techniques.

**Strengths:**

1. The paper tackles an important problem that aims to give guarantees on the generalization abilities of LLMs, which are getting more powerful but the good performance is extremely hard to interpret and assess.
2. Prior works compute non-vacuous bounds on LLMs but rely on assumptions of IID documents and therefore can only be applied to those that generate poor text quality. This work presents a novel approach based on properties of martingales and gives much tighter bounds on LLMs of much more practical capabilities. Further, it does not require altering the pretraining pipeline of the LLMs being analyzed.
3. It investigates the generalizations by examining compression schemes including Monarch matrices and Kronecker factorizations, and post-training quantization techniques. The results also give interesting insights for practitioners.

**Weaknesses:**

1. Since the utilization of martingales is one main theoretical contribution of the work, I feel some background and proof sketch on how they are being used would be better included in the main text.
2. The main models being examined are of the LLaMA and GPT-2 family of models. In particular, the experiment on the chat version of LLaMA is interesting as the generalization gets worse from the supervised finetuning. It would be interesting to see if this is generally true and why this is the case. More experiments on other finetuned LLMs would provide more evidence.
3. From table 2 and 3, it seems that given larger model sizes, the derived bounds get closer to random guess performance. Why is this the case and does it mean the bound would potentially be no-longer meaningful if the model gets large enough? Maybe I misunderstood something and would appreciate some clarification on this point.

**Questions:**

Stated in the prior part.

**Limitations:**

The authors note the limitations of the current work.

---

> ### Author Rebuttal · Authors · 2024-08-07
>
> We thank you for your thoughtful and supportive feedback! We respond to your questions below.
>
> **Effect of finetuning on downstream task performance:**
>
> As per your suggestion, we run additional experiments where we finetune the GPT2 large model (774M parameters) – pretrained on the WebText dataset – on multiple downstream tasks: grade school math, stack exchange and planning. These datasets are publicly available on HuggingFace. We report the results in the table below. While finetuning visibly improves the model’s performance on the downstream task, it negatively affects the performance on the pretraining dataset and hence leads to a worse upstream bits-per-dimension (BPD) bound since the compressed size of the model remains approximately constant.
>
> | Model | Compressed Size (MB) | Downstream Empirical Risk | Upstream Empirical Risk | Upstream BPD Bound  |
> |  ---    |               ---              |             ---             |                 ---              |               ---                   |
> | Pretrained GPT2 Large  | 424.07 | 3.44 (on planner); 4.56 (on stackexchange) ; 0.45 (on grad school math) | **4.65** | **10.47** |
> | GPT2 large finetuned on grade school math    | 420.65 | **0.016**  (on grad school math) | 6.92  | 12.72  |
> | GPT2 large finetuned on Stack Exchange      | 424.06 | **0.14**  (on stackexchange)   | 4.80 | 10.62 |
> | GPT2 large finetuned on Planning       | 424.07  | **0.001** (on planner) | 4.78  | 10.56  |
>
> **Trade-off between the empirical risk and the compressed size of the model:**
>
> Our generalization bounds, as described by Equation 2, can be conceptually written as:
>
> $$\text{Expected Risk} \leq \text{Empirical Risk} + \sqrt{\text{Compressed Model Size} / \text{Train Data Size}}.$$
>
> Therefore, the trade-off between the empirical risk and the compressed model size heavily depends on the compression approach and the size of the dataset. Having a dataset that contains a higher number of tokens puts a bigger emphasis on the empirical risk compared to the compressed model size, and vice versa. Likewise, if the compression approach is very aggressive, it will significantly reduce the compressed model size while potentially causing a deterioration of the empirical performance. The rate at which each element of the bound changes dictates whether the bound will increase or decrease for larger models. We demonstrate this effect in Figure 2 in the attached Rebuttal PDF, where we see that an aggressive compression scheme for the GPT2 models consisting of pretraining them in restricted SubLoRA spaces with 25k parameters only then quantizing them leads to an improvement in the token-level bounds as we increase the size of the original model. In contrast, quantizing the LLaMA2 models using QuIP# maintains a good empirical performance but does not reduce the compressed size significantly compared to an approach like SubLoRA, therefore the bounds deteriorate as the models become larger. If the Amber dataset contained more tokens, one would expect that the bounds would improve for larger models given the improvement in the empirical risk.
>
>
> **An intuitive application of our bounds:**
>
> In addition to the above experiments, we ran additional experiments for the antibody design downstream task to provide more intuition on the quantity that we bound .In fact, biologists propose mutations to existing antibody sequences by _changing the amino acid tokens at specific positions in the sequence_. Recent works have shown that LLMs pretrained on large antibody datasets can be used to propose mutations _conditioned on starting antibody sequences_. **Our token-level generalization bounds match the antibody design setting by bounding the expected next amino acid token negative log likelihood averaged over training contexts that serve as starting sequences for iterative mutations.** We provide results in the general response showing that these bounds are non-vacuous.
>
> **Background and proof sketch of the main theorem:**
>
> We provide a sketch of the proof in Appendix B.1, and it consists of three components: set up the empirical loss as the average of a martingale difference sequence, apply Azumas inequality for each hypothesis assigning failure probability proportional to the prior p(h), and then apply a union bound to relate the concentration around the many for individual hypotheses to any given hypothesis that can depend on the training data. We take your point that it would be beneficial to have this summary and some additional background in the main text, to provide additional context for the bounds we use. We will update the paper accordingly.
>
> We provide additional experiments in the general response showing that not only is the quantity we bound predictive of downstream performance, but the bounds themselves positively correlate with downstream performance.
>
> Thank you again for your supportive feedback. We made a significant effort to address your comments and run several new experiments inspired by your feedback, which we believe have improved our paper. We would appreciate it if you would consider raising your score in light of our response. We would be happy to engage in additional discussion if there are further questions.

---

> > ### Comment · Reviewer_iy2E · 2024-08-14
> >
> > Thank you for your response. I will maintain my score and recommend for acceptance.

---

### Author Rebuttal · Authors · 2024-08-07

We thank all the reviewers for their very supportive and helpful feedback. Inspired by the reviewers’ comments, we now report additional results that highlight the following contributions: (i) we complement our other understanding-oriented experiments through an antibody design setting where the task itself naturally matches the definition of the expected risk in our bounds since one would use LLMs pretrained on large antibody datasets to propose mutations **conditioned on starting antibody sequences from the training dataset**; (ii) we run additional experiments to further investigate the effect of finetuning models on their upstream performance and upstream generalization guarantees; (iii) we show that our bounds are predictive of downstream performance, achieving 98.9% and 99.4% correlation with downstream perplexity and error, respectively; (iv) we empirically demonstrate that the trade-off between the empirical risk and the compressed size of the model depends on the size of the dataset and the compression approach. The new figures and table can be found in the attached Rebuttal PDF.

We have also incorporated reviewer feedback to provide additional background on Martingale bounds as well as a proof sketch in the main text in the revised manuscript.

We begin with a general response and then address reviewers individually as separate posts.

**Intuitive interpretation of our bounds for antibody design:**

We provided several experiments in our submission that help interpret the bounds. We thought it would also be especially exciting to consider a downstream task beyond next-word prediction. We believe this is the first time generalization bounds for language models have been used in such a way — to guarantee downstream performance on a scientific problem.

Our token-level generalization bounds are particularly descriptive of antibody design in biology. An antibody sequence is usually composed of 20 different amino acid tokens to bind to a target of interest. An example of an antibody sequence from the Observed Antibody Space (OAS) database is the following: “SETLSLTCTVSGGSMSSY…” [1]. In therapeutic antibody design, biologists propose mutations to existing antibody sequences by changing the amino acid tokens at specific positions in the sequence. In our example sequence, a mutation is introduced if we change one or many amino acid tokens. The next-token prediction task in language modeling thus has a natural interpretation of predicting mutations at position i conditioned on positions <i. Recent works have shown that LLMs pretrained on large antibody datasets can be used to propose mutations conditioned on starting antibody sequences. **Our token-level generalization bounds match the antibody design setting by bounding the expected next amino acid token negative log likelihood averaged over training contexts that serve as starting sequences for iterative mutations.** In the table below, we show that language models based on the Mistral 7B architecture pretrained on a processed subset of the Observed Antibody Sequences (OAS) from scratch achieves non-vacuous token-level generalization bounds:

| Compression Approach|BPD Bound|Top-1 Error Bound|Validation Loss|
| ---| ---| --- | --- |
|Mistral 377M |2.41|31.60|0.28|
|Mistral 212M|2.06|26.25|0.30|
|Mistral 94M|**1.62**|**19.40**|0.30|
|Random Guess| 4.86| 96.56|1.46|

**Further investigating the effect of finetuning LLMs on upstream performance:**

In our work, we show that chat versions of the LLaMA models obtain worse bounds on the Amber dataset. We extend these experiments to GPT2 models finetuned for different purposes, namely to answer grade school math questions, coding questions, and to do planning. These downstream datasets are publicly available on HuggingFace. We report the results for pretrained GPT2 Large (774M) in Table 1 of the attached Rebuttal PDF. While finetuning visibly improves the model’s performance on the downstream task, it negatively affects the performance on the pretraining dataset and hence leads to a worse upstream bits-per-dimension (BPD) bound since the compressed size of the model remains approximately constant.

**Correlation between our bounds and the performance on downstream tasks:**

Following the suggestion of reviewer mAFv, we compute the correlation between our bounds and downstream performance. In particular, we compute token-level bounds for GPT2 small, medium, and large pretrained with SubLoRA with an intrinsic dimensionality of 25,000 and a LoRA rank of 4 on the OpenWebText dataset. In Figure 1(left y-axis) reported in the attached Rebuttal PDF, we plot the average zero-shot error (Error) and the perplexity (PPL) achieved by GPT2 models on downstream tasks, as reported in Table 6 of the original submission. On the right y-axis, we plot token-level bounds achieved by GPT2 on OpenWebText. Our token-level BPD bounds achieve **98.9%** and **99.4%** correlation with the downstream perplexity and error, respectively, and are indeed predictive of generalization on downstream tasks.

**Trade-off between the empirical risk and the compressed size of the model:**

We demonstrate in Figure 2 in the attached Rebuttal PDF that the trade-off between the empirical risk and the compressed model size heavily depends on the compression approach.

**Summary:**

We are thankful for the supportive feedback from the reviewers and believe that their input has made a positive impact on our paper. We make a significant contribution in our work by computing non-vacuous bounds at the LLaMA-70B scale and use our bounds to derive insights about generalization in LLMs, highlighting the remarkable ability of transformer models in capturing longer range correlations, and distinguishing between memorization and reasoning.
___
Reference:

[1] Olsen TH, Boyles F, Deane CM. Observed Antibody Space: A diverse database of cleaned, annotated, and translated unpaired and paired antibody sequences. Protein Science. 2022; 31: 141–146.

---

### Decision · Program_Chairs · 2024-09-25

**Decision:**

Accept (spotlight)

**Comment:**

The paper proposes a new approach for computing generalization bounds for large language models (LLMs). All the reviewers found the paper to be interesting, as it tackles an important problem. Meanwhile, the approach is novel as it "is a significant departure from the document-level bounds prevalent in prior works". There were some concerns on the clarity of the paper and also the lack of necessary background knowledge, while the authors did a good job on addressing that. So I would recommend accepting this paper.